# Photodynamic Opening of the Blood–Brain Barrier and the Meningeal Lymphatic System: The New Niche in Immunotherapy for Brain Tumors

**DOI:** 10.3390/pharmaceutics14122612

**Published:** 2022-11-26

**Authors:** Oxana Semyachkina-Glushkovskaya, Andrey Terskov, Alexander Khorovodov, Valeria Telnova, Inna Blokhina, Elena Saranceva, Jürgen Kurths

**Affiliations:** 1Department of Biology, Saratov State University, Astrakhanskaya Str. 83, 410012 Saratov, Russia; 2Institute of Physics, Humboldt University, Newtonstrasse 15, 12489 Berlin, Germany; 3Department of Complexity Science, Potsdam Institute for Climate Impact Research, Telegrafenberg A31, 14473 Potsdam, Germany

**Keywords:** photodynamic therapy, blood-brain barrier, meningeal lymphatic vessels, glioblastoma, immunotherapy for brain tumors

## Abstract

Photodynamic therapy (PDT) is a promising add-on therapy to the current standard of care for patients with glioblastoma (GBM). The traditional explanation of the anti-cancer PDT effects involves the PDT-induced generation of a singlet oxygen in the GBM cells, which causes tumor cell death and microvasculature collapse. Recently, new vascular mechanisms of PDT associated with opening of the blood–brain barrier (OBBB) and the activation of functions of the meningeal lymphatic vessels have been discovered. In this review, we highlight the emerging trends and future promises of immunotherapy for brain tumors and discuss PDT-OBBB as a new niche and an important informative platform for the development of innovative pharmacological strategies for the modulation of brain tumor immunity and the improvement of immunotherapy for GBM.

## 1. Photodynamic Therapy of Glioblastoma

Worldwide, brain tumors have been diagnosed in 300,000 people, among whom 250,000 patients have died [1,2]. In 2000, the 5- and 10-year survival rates for aggressive brain tumors were approximately 12% and 9%, respectively. Currently, this situation has improved, and the 5- and 10-year survival rates for malignant brain cancers are approximately 36% and 31%, respectively [3]. However, despite the improvements in therapy for brain tumors, the survival rate is still low compared to other types of cancer [1].

Glioblastoma (GBM) is the most common and aggressive forms of brain cancer [4,5]. GBM remains the one of deadliest of brain tumors. A guided surgical resection is the first step in treating patients with GBM [6,7]. However, even if GBM resection is successfully performed, 80% of GBMs recur within 2 cm of the original lesion [8,9,10]. There is no standard treatment for relapse. Surgery, radiation therapy, and chemotherapy or systemic therapy with bevacizumab are all options, depending on patient circumstances [11]. Typically, a patient who has surgery will receive chemotherapy, which is often combined with radiation [10]. However, despite all these treatments, the median survival time (MST) for most patients with GBM is less than one year, with a 5-year survival rate of less than 7% [12,13]. There are several challenges limiting the effective treatment of GBM [8]. First, patients can have an inoperable GBT due to tumor location, e.g., in the functional centers, which involve language, senses, vision and memory [14,15]. Second, the blood–brain barrier (BBB), due to its semipermeable properties, limits the delivery of a vast majority of cancer therapeutics, including monoclonal antibodies, antibody-drug conjugates, and hydrophilic molecules [16]. Third, GBM is characterized by heterogeneity at the cellular level, which induces different responses to anti-tumor pharmacological agents, leading to the failure of targeted therapies [17,18]. The challenge for GBM therapy is, therefore, to improve the control of tumor growth, especially for non-surgical patients.

Over the past 20 years, the innovative technology known as photodynamic therapy (PDT) has been developed for the treatment of GBM [19,20,21,22,23,24,25,26,27,28,29,30]. There are two modalities of PDT: intracavitary PDT and interstitial PDT (iPDT) (Figure 1). Intracavitary PDT is performed after the surgical resection of primary GBM with the use of an expandable balloon illumination device insufflated inside the surgical cavity [22,23]. The iPDT modality is a minimally invasive treatment for GBM without the large craniotomy that is used for non-surgical lesions or for therapy for recurrent GBM. iPDT is delivered with semiconductor lasers positioned inside the surgical cavity or directly in the tumor mass [20,24]. PDT combines a light source and nontoxic photosensitizing agents (photosensitizers, PSs). Systematically or topically administered PS is specifically accumulated in tumor tissues. Currently, the United States Food and Drug Administration (FDA) approved 5-aminolevulinic acid (5-ALA) for the fluorescence guided resection of GBMs and PDT of brain tumors [25]. 5-ALA exhibits high selectivity in terms of preferential accumulation in GBM tissues [26]. When the concentration of PS is sufficient, it is activated by exposure to light appropriated for its excitation (e.g., protoporphyrin IX (PpIX) induced by 5-ALA after its excitation with a 635 nm laser). The excited PS stimulates an energy transfer to endogenous oxygen (triplet state), leading to the generation of a singlet oxygen and resulting in tissue oxidation. In the singlet state, energy is converted to heat by internal conversion or emitted as fluorescence. In the triplet state, the energy generates the ROS-inducing damage within tumor cells by direct injury via necrosis and apoptosis or due to the occlusion of tumor vessels (thrombus formation) [19] (Figure 1). There are several clinical reports suggesting that PDT is a promising add-on therapy to the current standard of care for patients with GBM, especially in its recurrent and unresectable forms [19,20,21,22,23,24,25,26,27,28,29,30]. In the last 10 years, new PDT technologies have been developed based on the use of new-generation lasers capable of generating a singlet oxygen without PS [31,32,33]. PS-free-PDT may become a promising therapeutic approach in developing a breakthrough technology for the non-invasive treatment of GBM in children in whom PDT and radio- and chemotherapy are strongly limited, as well as in adult patients with allergic reactions to PSs. It should also be emphasized that children often show a superficial location of GBM in the brain [34,35], making the non-invasive treatment of pediatric tumors with PS-free-PDT possible.

The traditional explanation of the anti-cancer PDT effect is that the single^-^oxygen-induced damage of the endothelial cells results in tumor cell death and microvasculature collapse [19,20,21,27]. However, in recent studies, the new vascular effects of PDT associated with the opening of the BBB and the activation of lymphatic drainage of the brain tissues have been discovered [36,37,38,39,40,41,42,43,44]. In the next sections of this review, we discuss the mechanisms of PDT-OBBB and the PDT-OBBB-mediated modulation of brain tumor immunity (Figure 2).

## 2. Photodynamic Opening of the Blood–Brain Barrier

The BBB plays an important role in the regulation of CNS health and protects the CNS from toxins and pathogens [45]. Due to these properties, the BBB limits the delivery of a vast majority of anti-tumor therapeutics [15]. However, the question of whether the BBB actually prevents drug delivery into a GBT remains controversial [46]. It well known that the BBB is highly permeable in the mass of a GBM [47,48,49]. Nevertheless, the GBM cells spread via the blood vessels with an intact BBB [46,47]. Therefore, an effective GBM therapy depends on the development of new strategies to prevent the migration of the GBT cells via healthy cerebral vessels with an intact BBB [46,49,50,51].

In 2008, it was revealed that PDT, as an alternative method for GBM therapy, can also open the BBB [36,37,38,39,40,41,42,43,44]. Hirschberg was the first to publish the results of a study using inbred male Fischer 344 rats that showed that ALA-mediated PDT (400 µm bare flat-end quartz fiber (635 nm)) causes OBBB [36]. The BBB was opened 2 h after PDT treatment and completely restored 72 h after laser irradiation. No effect was observed on the BBB in the absence of 5-ALA. These findings clearly demonstrate PDT dose-related changes in rat brains. At the low fluence level of 9 J/cm^2^, 5-ALA-mediated PDT opened the BBB without any damages to brain tissues, with fast restoration of BBB permeability. At the high fluence level of 26 J/cm^2^, 5-ALA-mediated PDT caused both BBB disruption and damage to surrounding tissues (necrosis, edema, and hemorrhages), which were observed during the 2 weeks after 5-ALA-mediated PDT treatment.

In our experiments with 103 mongrel male mice, using different doses of laser radiation (635 nm, 10–40 J/cm^2^) and 5-ALA (20 and 80 mg/kg, i.v.), we revealed the optimal doses of PDT-OBBB (laser dose—15 J/cm^2^ and 5-ALA—20 mg/kg) for the fast recovery of the BBB after PDT treatment [42]. The higher doses of laser radiation or 5-ALA had no amplifying effects on BBB permeability, but they were associated with severe damage to the brain tissues. Zhang et al., using the method of skull optical clearing (USOCA [52]), showed non-invasive PDT-OBBB for an albumin complex of Evans Blue dye 68.5 kDa, TRITC dextran 70 kDa, and 100 nm GM_1_-liposomes in BALB/c mice [41]. Later, this group demonstrated that the PDT-OBBB-related levels of high weight molecules and solutes was more pronounced in 4-week-old mice than in 8-week-old BALB/c animals [40]. They also developed a technology for the quantitative and qualitative assessment of PDT-OBBB to Evans Blue based on spectral imaging through the optical clearing skull window [44].

Matsen et al. showed the efficiency of PDT-OBBB using macrophages loaded with gold-nanoparticles to improve the photothermal therapy (PTT) of GBM [39]. To study the efficiency of the PDT-OBBB, they used a photosensitizer, aluminium phthalocyanine disulfonate (AlPcS2a, 670 nm), along with a light dose of 2.5 J/cm^2^) and iron oxide nanoparticle (120–180 nm)-loaded exogenous macrophages as a contrast agents for the magnetic resonance imaging of the BBB’s permeability [37,38]. The accumulation of iron oxide-loaded macrophages was observed in both the MRI scans and histological sections of the non-tumor bearing rat brains following PDT-OBBB.

Trinidad et al., using a human FaDu cancer cell line (head and neck squamous cell carcinoma), demonstrated a significant synergy effect of combining PDT (AlPcS2a-PDT at a fluence level of 0.25 J/cm^2^) and PTT (14 W/cm^2^ irradiance) compared to applying each modality separately [53].

The mechanisms responsible for PDT-OBBB-induced changes remain poorly understood. In an experiment on mice, using ex vivo confocal microscopy, it was found that PDT-OBBB caused a temporal decrease in the intensity of the signals from the tight junctions (TJ), such as claudin-5 (CLDN-5) and VE-cadherin, as the main components of BBB integrity [41]. This could be explained by the temporal loss of TJ on the surface of the cerebral endothelium due to possible activation of the internalization of TJ [41]. Indeed, PDT-OBBB induced an elevation of the arrestin beta-1 (ARRB1) level in the brain tissues that is responsible for starting the internalization process [54]. It was shown in an in vitro model of the BBB that an increase in BBB permeability is associated with the ARRB2-mediated internalization of the VE-cadherin [55]. PDT has a direct effect on increasing the gaps of the TJs between the cerebral endothelial cells via changes of cytoskeleton and vascular tone loss due to microtubule depolarization [56,57]. One mechanism of PDT-OBBB may be an imbalance in the control of aquaporin water channels in the astrocytes [58,59,60,61]. This hypothesis is based on experimental results suggesting a relationship between PDT-OBBB and the development of vasogenic edema after the use of different photosensitizers [41,60,61]. Oxidative stress is another possible mechanism underlying PDT-OBBB, as evidenced by the increase in malondialdehyde levels in the mouse brains [41,62].

Thus, PDT-OBBB is an important and informative platform for better understanding the mechanisms underlying the cerebrovascular effects of fluorescence-guided resections of brain tumors and for improving the guidelines for the PDT treatment of GMB.

## 3. PDT-OBBB Modulation of Brain Tumor Immunity

Recently, it was discovered that GBM is characterized by the reduced outflow of cerebrospinal fluid and that the meningeal lymphatic vessels (MLVs) regulate brain tumor drainage and immunity [63,64,65,66,67]. This new knowledge about the functions of MLVs opens new strategies in the stimulation of efficient immune responses against gliomas [64,68]. Hu et al. showed that the dorsal MLVs underwent extensive remodeling in mice with intracranial glioma-grafted GL261 cells. They revealed changes in the gene that controlled MLV remodeling, fluid drainage, and inflammatory and immunological responses in mice with glioma [64]. Notably, the mice with glioma and an overexpression of vascular endothelial growth factor C (VEGF-C) displayed better responses to the anti-tumor therapies, including combinations of the anti- programmed death-1 protein (PD1) and cytotoxic T-lymphocyte-associated protein 4 (CTLA-4) blockade. However, the blockade of key lymphatic proteins abolished this effect [64]. Indeed, the mice with glioma that were treated with lymphatic protein blockers, such as chemokine (C-C motif) ligand 21 (CCL21) and C chemokine receptor type 7 (CCR7), did not demonstrate improvements after receiving this anti-tumor therapy. These pioneering data suggest that VEGF-C potentiates checkpoint therapy via the CCL21/CCR7 pathway. Song et al. discovered that VEGF receptor-3 (VEGFR3) enhanced immune surveillance from GBM and improved the effectiveness of anti-tumor therapies with checkpoints inhibitors [69]. Both Hu et al. [64] and Song et al. [69] documented an important role of the MLVs in the regulation of brain tumor immunity in mice. Thus, the brain local delivery of VEGF-C can improve the anti-tumor effects of immune checkpoint inhibitors and enhance the presence of tumor-infiltrating lymphocytes in the microenvironment of GBM [63,70,71,72].

OBBB is accompanied by the rapid activation of lymphatic drainage of the brain tissues and clearance of unnecessary molecules from the CNS [43,73,74,75,76]. Using two methods for OBBB—loud music or sound and PDT—we demonstrated that OBBB is associated with the activation of the lymphatic clearance of different tracers (Evans blue dye, dextran 70 kDa, beta-amyloid, and gold nanorods) that crossed OBBB (from the brain and its removal into the deep cervical lymph nodes (dcLNs)) [43,73,74,75,76]. The activation of the clearance of substances from the brain after OBBB can be explained by the hypothesis of Monro and Kellie [77]. According to their hypothesis, the cerebral spinal, extracellular, and intestinal fluids create a state of volume equilibrium. Therefore, any increase in the volume of one of these fluids should be compensated by a brain drainage, i.e., the removal of fluids from the brain. OBBB causes a fluid influx from the bloodstream into the perivascular spaces that, in the case of PDT-OBBB, can induce vasogenic edema [60,61]. We hypothesized that OBBB stimulates the drainage of the brain tissues in a way that maintains extracellular homeostasis.

The study of the photo-modulation of MLVs and the mechanisms underlying this phenomenon is still in its infancy. The changes in the MLVs associated with OBBB may be explained by the expression of a number of similar proteins in the endothelium of lymphatic and blood vessels. Indeed, the lymphatic vessels are composed of the TJ proteins typical for the blood endothelium, such as claudin-5, a family of ZO proteins, and endothelial selective adhesion molecule; VE-cadherin; junctional adhesion molecule (JAM); and PECAM-1/CD31 [78,79]. In various series of our experimental studies, we showed that PDT-OBBB caused temporal changes in TJs expression in both the blood and lymphatic endothelium [41,75,80].

The PDT-mediated activation of lymphatic functions can be also explained by a PTD-induced activation of endothelial nitric oxide (NO) synthase [81]. The NO causes the relaxation of the blood and lymphatic endothelium via the opening of calcium (Ca^2+^)-activated potassium channels that lead to the dilation of the blood and lymphatic vessels [82]. In series of experiments using the PS-free generation of a singlet oxygen by a 1267 nm laser, it was discovered an important role of NO in the photo-induced dilation of the MLVs led to an increase in the lymphatic evacuation of red blood cells (RBCs) from mouse brains with intraventricular hemorrhages [83]. The non-specific blockade of the NO synthases inhibited the photo-mediated relaxation of the MLVs and the evacuation of RBCs from the brain into the dcLNs [83]. We assumed that PDT’s effects on the MLVs facilitated the lymphatic drainage of the brain tissues and the removal of cells/macromolecules from the CNS to the dcLNs.

Another mechanism of the singlet-oxygen-mediated control of vascular tone is the ability of a singlet oxygen to regulate endothelial relaxation [84]. The singlet oxygen induces the oxidation of the amino acid tryptophan in mammalian tissues, which leads to the cellular production of metabolites, such as *N*-formylkynurenine, with the activation of the haem-containing enzyme indoleamine 2,3-dioxygenase 1. This enzyme is widely expressed in the blood and lymphatic endothelium and contributes to the relaxation of vascular tone [84,85,86]. It was recently discovered that endothelial indoleamine 2,3-dioxygenase 1 caused the dilation of blood vessels via the formation of a singlet oxygen [84]. These pioneering findings shed light on the roles of singlet oxygen in the regulation of vascular tone, opening new perspectives on the modulation of systemic inflammatory responses of vasculature.

PDT has several unique properties that induce immunogenic cell death (ICD) [87,88,89,90,91,92,93,94,95,96,97]. Among the different mechanisms of PDT-induced ICD is the damage-associated molecular patterns (DAMPs) released from tumor cells caused by PDT [98,99,100]. In turn, DAMPs mediate the formation of antigen-presenting cells (APCs). The PDT activation of APCs causes their migration and proliferation in local lymph nodes where the APCs then present tumor antigens to th CD8+ T cells [87]. The MLVs are pathways for APC traffic from the CNS into the dcLNs [63,69,70,71]. The activated CD8+ T cells induce the apoptosis of tumor cells, thereby providing long-term tumor control. Therefore, PDT-induced ICD has the potential to stimulate immune activation, contributing to long-term tumor control [87,88,89,90,91,92,93,94,95,96,97]. However, the efficiency of PDT-induced ICD is reduced in a hypoxic tumor microenvironment, which limits the antitumor effects during PDT-related immunostimulation [101]. There is an emerging trend of using PDT-induced ICD in combination with nanotechnology to overcome this limitation [88,89,93,94,97]. The growing body of evidence indicates that manganese dioxide and catalase can stimulate the production of O_2_ via the catalysis of hydrogen peroxidase [102,103]. Therefore, nanocarries loading with manganese dioxide, catalase, and other oxygen-boosted PDTs are considered to increase tumor ICD [88,94,102,103]. Furthermore, perfuorocarbons and hemoglobin are used for the direct delivery of O_2_ in tumor hypoxic microenvironments [94]. In addition, another limitation of PDT is the low level of ICD induced by PDT, which is not enough for effective antitumor effects [97]. To solve this problem, a combination of PDT-induced ICD with PTT and chemotherapy has been proposed to increase the effectiveness of tumor ICD [94]. Currently, the control of the escape of immune cells into the tumor microenvironment using inhibitors of immune checkpoints is considered as a most promising strategy in antitumor therapies [104,105,106,107,108,109]. The luck of signaling molecules in tumors that can activate the generation of T cells in the lymphoid organs is one reason that only few numbers of effector T cells can enrich the tumor region, which allows tumor cells to evade immune surveillance [104,105]. Moreover, checkpoints generally inhibit the activated T cells after antigen recognition, leading to a decrease in antitumor immunity [105,106,107,108]. Therefore, in the past decade, the use of immune checkpoint blockers, such as CTLA4, PD-1, programmed death-ligand 1, and indoleamine 2,3-dioxygenase, has been proposed for the improvement of antitumor therapies [105,106,107,108,109,110,111,112]. The combined antitumor therapy of PDT-induced ICD and inhibitors of immune checkpoints is expected to be a new and promising strategy in tumor immunotherapy, including GBM treatment [87,94].

PDT-treated GBM cells may also act as a therapeutic anti-tumor vaccine [87,113]. In earlier studies, Etminan et al. demonstrated that the ALA-PDT of GBM cells promoted the attraction of dendritic cells and stimulated their maturation [114]. Indeed, co-cultured ALA-PDT-treated GBM cells with DCs induced the expression of CD_83_, a marker for the maturation of DCs, as well as the expression of stimulatory factors, such as CD_40_, CD_80_, and CD_86_. Photofrine-PDT-treated C6 glioma cells co-cultured with dendritic cells caused more effective DC differentiation, with a high expression of CD_80_ and major histocompatibility complex II [115]. The combination of Hypericin-mediated PDT with DCs in a high-grade glioma mouse model demonstrated the significant suppression of the growth of glioma cells [116]. An in vivo brain tumor model using F_98_ glioma cells has shown a systemic immune response generated by disulfonated aluminum phtalocyanine-mediated PDT co-cultured with macrophages [113]. Taking into account the limitations of light penetration into the brain, the further detailed studies of the effectiveness of PDT-produced vaccines for GBM in humans appears to be the most attractive approach. A better understanding of the activation of the antitumor immunity induced by PDT-produced vaccines would allow researchers to define the optimal protocols of the stimulation of the immune system to recognize and prevent the recurrence of GBM after surgery. The combination of PDT-induced anti-tumor vaccines with check-point inhibitors is an exciting field to explore.

## 4. Limitations of PDT in Therapy of GBT

PDT is a treatment modality which consists of light, oxygen, and PSs, and all these components have limitations [117,118,119,120]. The penetration of light depends on the optical properties of the head and brain tissues and the laser wavelength used for the PDT (Figure 3). There exists a heterogeneity between and within a tissue, leading to light scattering and a significant loss in the light energy that can reach the brain tissues. The brain fluids absorb light at longer wavelengths, and endogenous dyes, such as hemoglobin and melanin, absorb light at shorter wavelengths, which changes the light penetration depth into the brain tissues. Therefore, the photo-therapeutic window is between 600 and 1300 nm [121]. The light within a range of wavelengths between 620 and 850 nm has the most penetrating capability to achieve the maximum skin permeability [121,122,123] (Table 1). It is believed that light above 850 nm cannot generate a sufficient energy transfer to its triplet state to produce a singlet oxygen. However, in recent studies, it has been discovered that a 1267 nm light can generate singlet oxygen directly in different tissues, including the brain [31,32,33,124,125,126,127]. This discovery can open new horizons for developing PSs-free photo-therapies for GBM for patients that cannot use PDT due to allergies, age, and other limitations.

The ^1^O_2_ oxygen is another fundamental element of PDT. Indeed, the therapeutic efficacy of PDT strongly depends on the ^1^O_2_ content in the GBM cells. On the other hand, the GBM tissues are deprived of ^1^O_2_ as a result of the rapid growth of GBM cells and insufficient vasculature [128]. Furthermore, photosensitization itself quickly depletes cellular ^1^O_2_ levels in tumor cells. Therefore, the light dose has to be carefully adjusted and the light should be introduced in pulses [129]. For this reason, PDT is considered a self-limiting modality which causes its own inhibition. In order to overcome this problem, oxygen-boosted PDT is considered to increase the efficacy of PDT for the treatment of GBM [88,94,102,103].

PS is a key component of PDT. The PS accumulated in tumor cells is activated by a light with specific intensity and wavelength. The activated PS performs a photodynamic response, which underlies the anti-tumor effects of PDT. 5-ALA is the most widely used agent for the fluorescence-guided surgery of GBM, with further PDT the edge of the tumor [130]. It is generally accepted that 5-ALA accumulates directly in the GBM cells [26,131,132]. However, as discussed above, PDT with 5-ALA affects also the normal cerebral vessels, inducing OBBB in the different regions of the brain, which may be one reason for the vasogenic edema observed after 5-ALA-mediated PDT [36,37,38,39,40,41]. This provides a basis for revisiting our knowledge of the vascular effects of 5-ALA-mediated PDT for the optimization of the photo-diagnosis of and therapies for GBM.

**Table 1 pharmaceutics-14-02612-t001:** Depth of the penetration of light into the head and brain tissues in humans.

First Author, Year	Tissues, mm/Laser, nm	Penetration Depth (mm) and Transmittance (%)
Wan et al., 1981 [133]	Scalp + Skull, 9–13 mm	
	400 nm	10^−5^–10^−4^%
	546 nm	2 × 10^−4^–8 × 10^−4^%
	630 nm	2 × 10^−3^–9 × 10^−3^%
	664 nm	2 × 10^−2^%
	703 nm	3–10^−2^%
	856 nm	2 × 10^−2^–8 × 10^−2^%
Lychagov et al., 2006 [134]	Skull, 4–14 mm, 810 nm	1–16%
	Scalp and skull, 7–20 mm, 810 nm	0.5–5%
Svaasand and Ellingsen, 1983 [135]	♂ Neonatal brain	
	♀ Adult brain	
	488 nm	♂ 1.3 mm ♀ 0.4 mm
	514 nm	♂ 1.1 mm ♀ 0.4 mm
	660 nm	♂ 3.7 mm ♀ 1.2 mm
	1060 nm	♂ 7.1 mm ♀ 3.2 mm
Stolik et al., 2000 [136]	Adult brain	
	632 nm	0.92 mm
	675 nm	1.38 mm
	780 nm	2.17 mm
	835 nm	2.52 mm
Jagdeo et al., 2012 [137]	Skull 5 mm	
	633 mm	Left parietal lobe 3%
		Right parietal lobe 3.7%
		Frontal lobe 1%
	830 mm	Left parietal lobe 9%
		Right parietal lobe 9.2%
		Frontal lobe 5.9%
Yue et al., 2015 [138]	Scalp, skull, and adult brain (total 60 mm)	
	850 nm	Parietal, occipital, temporal, and frontal lobes, 10^−4^–10^−3^%
Barett and Gonzalez-Lima, 2013 [139]	Skull	
	1064 nm	Frontal lobe 2%

On the whole, the choice of optimal combinations of PS dose, light sources, and treatment parameters is very important in order to achieve successful results in the PDT of GBM. The emerging evidence confirms that 5-ALA-mediated PDT is effective not only in the field of brain oncology, but also in many other areas of medicine where surgery is impossible [117]. It is expected that the coming years will be a time of observation of the development of new diagnostic and therapeutic strategies with the use of 5-ALA acid. In addition, PDT-OBBB as new method of brain drug delivery deserves greater capacity and investment in clinical trials, which would allow scientists to learn more about the true translational potential of PDT.

## 5. Conclusions

Recently, it has been discovered that PDT, as a promising method for the prevention of the post-operation recurrence of GBM, causes temporal OBBB. The PDT-OBBB sheds light on the innovative strategies for the modulation of brain tumor immunity (Figure 4). The new knowledge about PDT-OBBB associated with the activation of lymphatic drainage and clearance of the brain tissues opens a new niche in immunotherapy for brain tumors. PDT is used as a promising add-on therapy to the current standard of care for patients with GBM, especially in its recurrent and inoperable forms. PDT-OBBB activates the MLV functions, which are pathways of traffic for the APCs produced by GBM after PDT. Therefore, PDT-OBBB can potentially stimulate the lymphatic traffic of the APCs into the dcLNs where the APCs activate the tumor antigens to the CD4+/CD8+ T cells. The T-cells migrate from the dcLNs to the GBM and extravasate in the tumor tissues via OBBB. Thus, PDT-OBBB can facilitate an increase in the number of infiltrating lymphocytes and T-cells, which induces GBM cytotoxicity, releasing more GBM antigens and contributing to brain tumor immunity. The immunotherapy with VEGF-C can be significantly improved by PDT-OBBB via contributing to the delivery of VEGF-C into the GBM. Thus, PDT-OBBB may promote the effectiveness of immune checkpoint inhibitors which have been shown to enhance the number and function of tumor-infiltrating lymphocytes in GBM patients. Therefore, PDT-OBBB is a new and important informative platform for the development of innovative pharmacological strategies for the modulation of brain tumor immunity and for the improvement of immunotherapy in brain tumors.

## Figures and Tables

**Figure 1 pharmaceutics-14-02612-f001:**
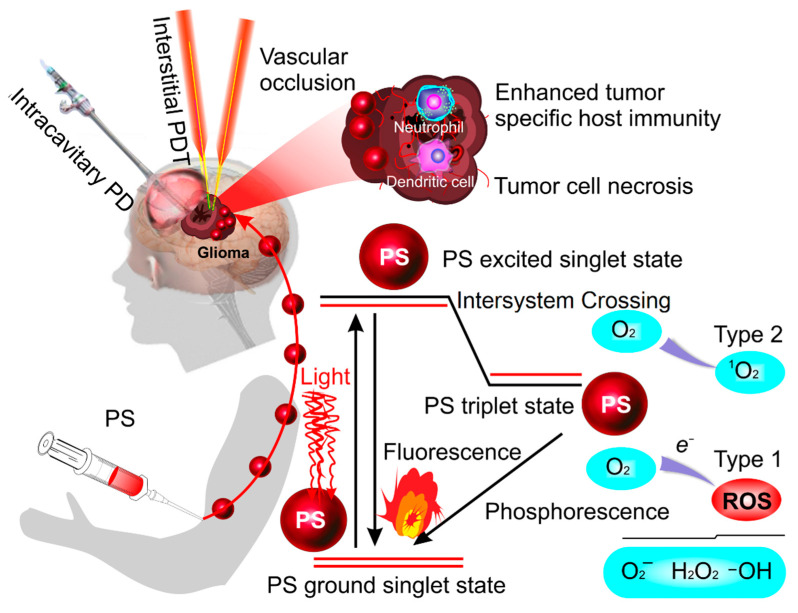
Schematic illustration of intracavitary PDT and interstitial PDT for GBM with an energy diagram of the oxygen response after PS excitation by an appropriate light wavelength. In the singlet state, energy is converted to heat by internal conversion or emitted as fluorescence. Then, PS in the excited singlet state can convert to the excited triplet state. In the triplet state, the energy can undergo a Type 1 or Type 2 redox reaction with the generation of the reactive oxygen species (ROS), which induces the death of the GBM cells via necrosis and/or apoptosis. This can also destruct the GBM vasculature and produce an acute inflammatory response that attracts leukocyte activation.

**Figure 2 pharmaceutics-14-02612-f002:**
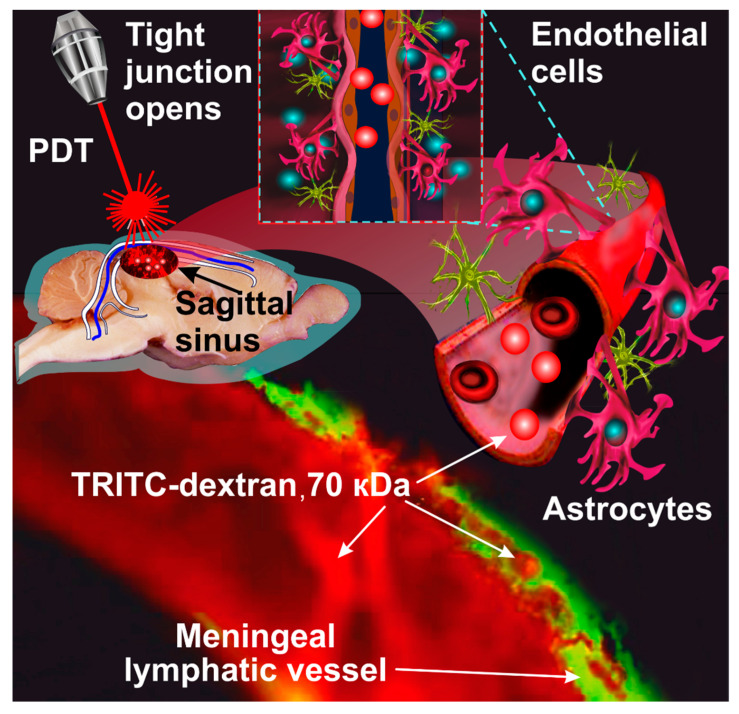
The lymphatic clearance of tracer from the brain after crossing the OBBB. The figure illustrates the fluorescent microscopy of the clearance of TRITC-dextran 70 kDa (red color) from the brain via the meningeal lymphatics (green color, labelled by the specific antibodies LYVE-1 conjugated with Alexa 488). The intravenously injected TRITC-dextran is immediately observed in the Sagittal sinus (the main cerebral vein), and 30 min later, it is also present in the meningeal lymphatics due to the PDT-OBBB-mediated activation of the lymphatic clearance of tracer from the brain [43].

**Figure 3 pharmaceutics-14-02612-f003:**
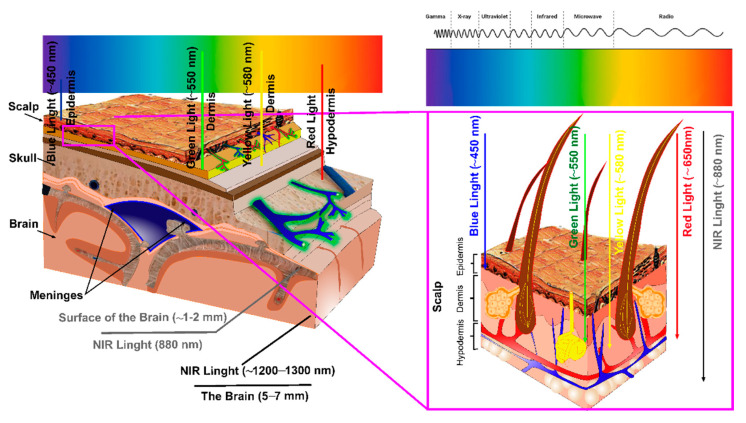
Schematic illustration of the depth of the penetration of light into the head and brain tissues, depending on the wavelength.

**Figure 4 pharmaceutics-14-02612-f004:**
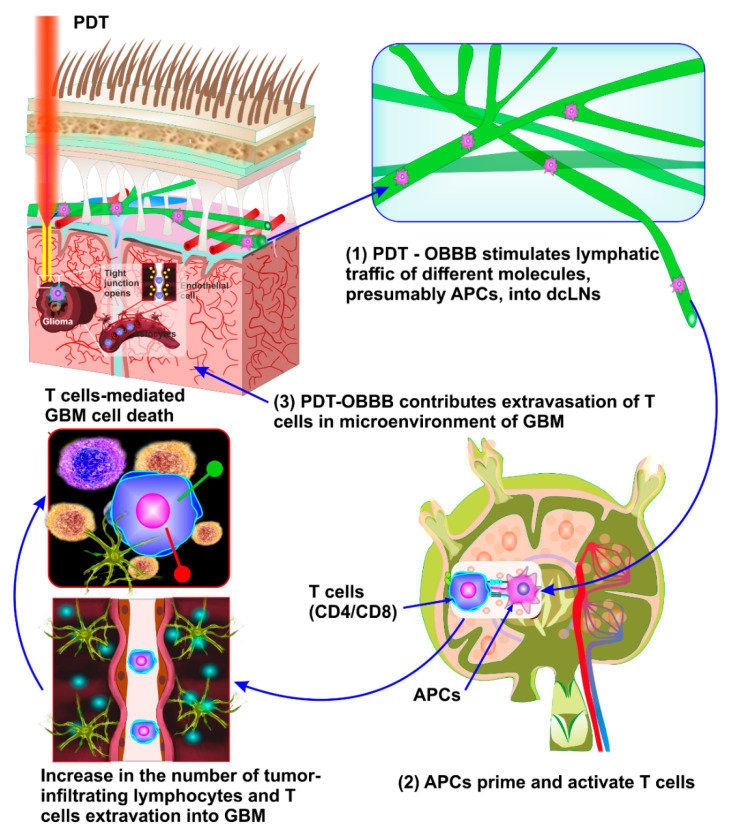
PDT-OBBB modulation of brain tumor immunity. The cancer immunity cycle in GBM involves the APC release and lymphatic traffic of APCs into the dCLNs (1), where they activate CD4+/CD8+-T cells (2). The T-cells migrate from the dcLNs to the GBM, extravasate into the tumor tissues via OBBB (3), and, finally, the T cells induce GBM cytotoxicity, releasing more GBM antigens and contributing to brain tumor immunity.

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
