# Peer review of "Photodynamic Opening of the Blood–Brain Barrier and the Meningeal Lymphatic System: The New Niche in Immunotherapy for Brain Tumors"

_pharmaceutics, 2022, doi:10.3390/pharmaceutics14122612_

Round 1

Reviewer 1 Report

The manuscript describes the important problem in the development of anti-glioma therapies. There were cited up-to-date publications in the manuscript. Authors described all data meticulously and carefully. I don't have any merit remarks. I have found only minor errors: lines 105, 299 and 300-there are “sessions”, I think it should be “sections”. From my point of view this paper definitely will very valuable both for basic scientists and clinicians.

Author Response

The manuscript describes the important problem in the development of anti-glioma therapies. There were cited up-to-date publications in the manuscript. Authors described all data meticulously and carefully. I don't have any merit remarks. I have found only minor errors: lines 105, 299 and 300-there are “sessions”, I think it should be “sections”. From my point of view this paper definitely will very valuable both for basic scientists and clinicians.

Response: The authors would like to thank the referee for the positive assessment of our review article and for important comments. We made corrections of the grammatical errors.

Let us once again thank the referee for the great help in improving our article.

Authors

Reviewer 2 Report

Dear Sir,

The review is of interest, the english is readable  but not in all the text.  Several paragraphs or sentences  must be reconsidered before publication (misreading).

Best regards

Author Response

The review is of interest, the English is readable but not in all the text. Several paragraphs or sentences must be reconsidered before publication (misreading).

Response: The authors express their sincere gratitude for the great help of the referee in improving our review article and for the valuable advices. The abbreviation list is included in the article. We made corrections the grammar errors. All changes in the text are highlighted in yellow.

Many thanks again for your interest in our article and for its positive evaluation.

Authors

Reviewer 3 Report

In this manuscript, the work by Semyachkina-Glushkovskaya et al. highlighted the emerging trends and future promises of immunotherapy for brain tumor and discussed that the photogynamic therapy associated with opening of the blood-brain barrier (PDT-OBBB) is the new niche and important informative platform for the development of innovative pharmacological strategies of modulation of brain tumor immunity and improvement of immunotherapy for glioblastoma (GBM). The article proposed that PDT-OBBB activates meningeal lymphatic vessels (MLVs), which in turn promoted the antigen presentation by APCs, activation and infiltration of CD4+/8+ T cells in the microenvironment of GBM. This is a relatively novel perspective and viewpoint, but it is not informative enough, while there are some areas that are not clearly explained and don’t meet the requirements of this journal very well. To name a few,

1The first point of the article is all about PDT for GBM, is this section too much? The focus of the article should be on the relationship between PDT and the immunotherapy of GBM

2Are there any limitations to the usefulness of photodynamic therapy in gliomas? For example, the intensity of the laser can be reduced or even be unable to penetrate due to the presence of the skull. In addition, the article mentions that excessive doses of photosensitizers can cause severe damage to brain tissue, etc.

3The mechanism of PDT-OBBB activation of MLVs needs to be further investigated and discussed, and it seems that the reduction of tight junction proteins is not convincing enough.

4Is it possible to give more examples of PDT causing ICD followed by activation of immune cells in the brain for the immunotherapy of GBM? In other words, there are not sufficient references cited in this area.

5In general, only the role of activated MLVs has been discussed in the article about the connection between PDT and the immunotherapy of GBM. Are there other aspects or perspectives to further elucidate?

Author Response

In this manuscript, the work by Semyachkina-Glushkovskaya et al. highlighted the emerging trends and future promises of immunotherapy for brain tumor and discussed that the photogynamic therapy associated with opening of the blood-brain barrier (PDT-OBBB) is the new niche and important informative platform for the development of innovative pharmacological strategies of modulation of brain tumor immunity and improvement of immunotherapy for glioblastoma (GBM). The article proposed that PDT-OBBB activates meningeal lymphatic vessels (MLVs), which in turn promoted the antigen presentation by APCs, activation and infiltration of CD4+/8+ T cells in the microenvironment of GBM. This is a relatively novel perspective and viewpoint, but it is not informative enough, while there are some areas that are not clearly explained and don’t meet the requirements of this journal very well. To name a few,

Comment: The first point of the article is all about PDT for GBM, is this section too much? The focus of the article should be on the relationship between PDT and the immunotherapy of GBM.

Response: The authors would like to express their sincere gratitude to the referee for constructive advices and valuable recommendations, which helped to significantly improve our review article.

We have shortened the section “Photodynamic therapy of glioblastoma” and expanded the section “PDT-OBBB-modulation of brain tumor immunity” (Lines 247-288).

Comment: Are there any limitations to the usefulness of photodynamic therapy in gliomas? For example, the intensity of the laser can be reduced or even be unable to penetrate due to the presence of the skull. In addition, the article mentions that excessive doses of photosensitizers can cause severe damage to brain tissue, etc.

Response: Thank you for this important remark. We added the section “Limitations of PDT in therapy of GBT” (Lines 294-355, Figure 3, Table 1).

Comment: The mechanism of PDT-OBBB activation of MLVs needs to be further investigated and discussed, and it seems that the reduction of tight junction proteins is not convincing enough.

Response: The study of photo-modulation of MLVs and mechanisms underlying this phenomenon is still in its infancy. Therefore, the literature containing information about the mechanisms of PDT-OBBB activation of MLVs is limited. We added the discussion of an important role of NO in photo-induced dilation of the MLVs and the mechanism of singlet oxygen-mediated control of the vascular tone under inflammatory conditions (Lines 198-232).

Comment: Is it possible to give more examples of PDT causing ICD followed by activation of immune cells in the brain for the immunotherapy of GBM? In other words, there are not sufficient references cited in this area.

Response: We added more information about the new promising strategies in GBT immunotherapy and the efficiency of PDT-related immunostimulation in the treatment of GBT (Lines 247-270).

Comment: In general, only the role of activated MLVs has been discussed in the article about the connection between PDT and the immunotherapy of GBM. Are there other aspects or perspectives to further elucidate?

Response: We added additional discussion about PDT-produced vaccines for GBM and prospect for the use of combination of PDT-induced anti-tumor vaccines with other therapeutic modalities including check-point inhibitors (Lines 271-288).

Let us thank you once again for the opportunity to improve the quality of our review article with your helpful comments and advices.

Authors

Reviewer 4 Report

1. Extracellular matrix of Brain is known to be composed of 40 % hyaluronic acid while normal tissues are not. Due to these properties, light penetration depth must be relatively  changed. 

2. In relation to comment 1. According to kinds of photosensitizers and light souce (wavelength, light power, etc....), the penetration depth and/or delivery capacity of photosensitizer against brain/GBM must be changed. If possible, would you study references and then presented as a Table ?

Author Response

  1. Extracellular matrix of Brain is known to be composed of 40 % hyaluronic acid while normal tissues are not. Due to these properties, light penetration depth must be relatively changed.
  2. In relation to comment 1. According to kinds of photosensitizers and light souce (wavelength, light power, etc....), the penetration depth and/or delivery capacity of photosensitizer against brain/GBM must be changed. If possible, would you study references and then presented as a Table?

Response: The authors express their deep gratitude for the opportunity to improve our review article with the help of the valuable advice of the reviewer. We added Figure 3 “Schematic illustration of depth of penetration of light into the head and brain tissues de-pending on the wavelength” and Table 1 “Depth of penetration of light into the head and brain tissues in humans” (Pages 8 and 9).

Let us thank you once again for reviewing our article and helping in its possible publication in Pharmaceutics.

Reviewer 5 Report

Semyachkina-Glushkovskaya and co-worker present in their submission to Pharmaceutics "Photodynamic opening of the blood-brain barrier and the meningeal lymphatic system: the new niche in immunotherapy for brain tumors". The following issues must be carefully fixed in the revision, which must be checked again.

There are no numbers for the keywords.

Figure 1: The transition from the S1 state to the T1 state is not called Relaxation but Intersystem Crossing (ISC).

Figure 2: Inside the Figure, it is written: kDa, instead of kDa.

"The BBB was opened 2 hours after PDT and completely restored 72 hours later if PDT." The English of this sentence must be improved.

"At the low fluence level of 9 J, ALA-PDT opens the BBB without any damages of brain tissues with fast restoration of the BBB penetration. At the high fluence level of 26 J," The unit "J" must be replaced with "J/cm2".

Same comment as above to "aluminum phthalocyanine disulfonate - 153 AlPcS2a, λ ¼ 670 nm; light dose = 2.5 J" What is the meaning of "λ ¼ 670 nm"? Correct the typo "aluminum" to "aluminium".

"Trinidad et al. using human FaDu cancer cell line (head and neck squamous cell carcinoma) demonstrate a significant synergy effect of combined PDT (AlPcS2a-PDT at a fluence level of 0.25 J/cm2) and PTT (14 W/cm2 irradiance) compared to each modality applied separately." The applied fluences in PTT and PDT should both be given in J/cm2.

"The activation of lymphatic functions after PDT can be also associated with a PTD-related increase" should be "The activation of lymphatic functions after PDT can be also associated with a PDT-related increase"

Ca2+: The "2+" should be superscript (several times).

References:

The article number should not be in brackets. The issue numbers are sometimes mentioned and sometimes not. The journal titles should all be abbreviated, see e. g. 33.

Ref. 3: The page numbers are v1-v100.

Ref. 8: The page numbers should be S2–S8.

Ref. 33: Sci. Rep. does not have issue numbers.

Ref. 34: The article number instead of the page numbers should be given.

Ref. 43: Page numbers are missing.

Ref. 69: As this is an early version, the DOI number should be given.

Ref. 89: The journal name is incomplete.

Author Response

Semyachkina-Glushkovskaya and co-worker present in their submission to Pharmaceutics "Photodynamic opening of the blood-brain barrier and the meningeal lymphatic system: the new niche in immunotherapy for brain tumors". The following issues must be carefully fixed in the revision, which must be checked again.

There are no numbers for the keywords.

Comments: Figure 1: The transition from the S1 state to the T1 state is not called Relaxation but Intersystem Crossing (ISC). Figure 2: Inside the Figure, it is written: kDa, instead of kDa.

Response: The authors would like to express their deep gratitude to the referee for valuable comments and constructive advices, which allowed to significantly improve our review article. We made changes in Figures 1 and 2 in accordance with the comments (Pages 2 and 3).

Comment: The BBB was opened 2 hours after PDT and completely restored 72 hours later if PDT." The English of this sentence must be improved.

Response: We made correction of this sentence (Lines 110, 111).

Comment: At the low fluence level of 9 J, ALA-PDT opens the BBB without any damages of brain tissues with fast restoration of the BBB penetration. At the high fluence level of 26 J," The unit "J" must be replaced with "J/cm2".

Response: We made correction of these sentences (Lines 112-117).

Comment: Same comment as above to "aluminum phthalocyanine disulfonate - 153 AlPcS2a, λ ¼ 670 nm; light dose = 2.5 J" What is the meaning of "λ ¼ 670 nm"? Correct the typo "aluminum" to "aluminium".

Response: We made correction of sentence (Lines 131-135).

Comment: Trinidad et al. using human FaDu cancer cell line (head and neck squamous cell carcinoma) demonstrate a significant synergy effect of combined PDT (AlPcS2a-PDT at a fluence level of 0.25 J/cm2) and PTT (14 W/cm2 irradiance) compared to each modality applied separately." The applied fluences in PTT and PDT should both be given in J/cm2.

Response: We made correction of sentence (Lines 138-140).

"The activation of lymphatic functions after PDT can be also associated with a PTD-related increase" should be "The activation of lymphatic functions after PDT can be also associated with a PDT-related increase"

Comment: Ca2+: The "2+" should be superscript (several times).

Response: We made correction of Ca2+ designation (Line 211).

Comments:

References:

The article number should not be in brackets. The issue numbers are sometimes mentioned and sometimes not. The journal titles should all be abbreviated, see e. g. 33.

Ref. 3: The page numbers are v1-v100.

Ref. 8: The page numbers should be S2–S8.

Ref. 33: Sci. Rep. does not have issue numbers.

Ref. 34: The article number instead of the page numbers should be given.

Ref. 43: Page numbers are missing.

Ref. 69: As this is an early version, the DOI number should be given.

Ref. 89: The journal name is incomplete.

Response: We made corrections of the references.

We thank the referee again for the great help in improvement of our review article and its possible publication in Pharmaceutics.

Authors

Round 2

Reviewer 2 Report

 Dear,

Only minor points to be corrected

Best regards

Reviewer 3 Report

I have no further concern